# Choline Metabolism to the Proatherogenic Metabolite Trimethylamine Occurs Primarily in the Distal Colon Microbiome In Vitro

**DOI:** 10.3390/metabo15080552

**Published:** 2025-08-16

**Authors:** Anthony M. Buckley, Sarah Zaidan, Michael G. Sweet, Duncan J. Ewin, Juanita G. Ratliff, Aliyah Alkazemi, William Davis Birch, Ashley M. McAmis, Andrew P. Neilson

**Affiliations:** 1Microbiome and Nutritional Science Group, Faculty of Food Science and Nutrition, School of Food Science, University of Leeds, Leeds LS2 9JT, UK; a.buckley1@leeds.ac.uk (A.M.B.); aliyahhosain4609@gmail.com (A.A.); 2Plants for Human Health Institute, North Carolina State University, 600 Laureate Way, Kannapolis, NC 28081, USA; sweetm197@gmail.com (M.G.S.); jgratlif@ncsu.edu (J.G.R.); ammcamis@ncsu.edu (A.M.M.); 3Faculty of Engineering and Physical Sciences, School of Mechanical Engineering, University of Leeds, Leeds LS2 9JT, UK; w.a.davisbirch@leeds.ac.uk; 4Department of Food, Bioprocessing, and Nutrition Sciences, North Carolina State University, Schaub Hall, Campus Box 7624, 400 Dan Allen Drive, Raleigh, NC 27606, USA

**Keywords:** gut microbiome, choline, trimethylamine, trimethylamine N-oxide *cutC*, TMA lyase

## Abstract

Background/Objectives: Gut microbial metabolism of choline and related quaternary amines to trimethylamine (TMA) is the first step in the production of trimethylamine N-oxide (TMAO), a circulating metabolite that contributes to the development of atherosclerosis and other forms of cardiovascular disease (CVD). No data exist on regional differences in TMA production within the colon due to difficulties studying gut regions in vivo. A better understanding of TMA production by gut microbiota is needed to develop strategies to limit TMA production in the gut and TMAO levels in circulation with the goal of reducing CVD risk. Methods: We employed our novel three-compartment MiGut in vitro model, which establishes three distinct microbial ecologies mimicking the proximal, mid, and distal colon, to study conversion of choline to TMA by human gut microbiota using isotopically labelled substrate. Results: Choline-d_9_ was almost completely converted to TMA-d_9_ in vessels 2–3 (mimicking the mid and distal colon) within 6–8 h, but little conversion occurred in vessel 1 (mimicking the proximal colon). Abundance of *cutC*, part of the *cutC/D* gene cluster responsible for choline conversion to TMA, was highest in vessel 1 vs. 2–3, suggesting that its expression or activity may be suppressed in the proximal colon. Another possibility is that the viability/activity of bacteria expressing *cutC* could be suppressed in the same region. Conclusions: This novel finding suggests that while bacteria capable of converting choline to TMA exist throughout the colon, their activity may be different in distinct colon regions. The regional specificity of TMA production, if confirmed in vivo, has implications for both basic microbial ecology related to CVD and the development of strategies to control TMA and TMAO production, with the goal of lowering CVD risk. These findings warrant further study in vitro and in vivo.

## 1. Introduction

Cardiovascular disease (CVD) is a global public health crisis [1] and is the leading cause of death worldwide [2,3]. Recently, interest has grown in the epidemiological and mechanistic relationships between the gut microbiome and CVD [4,5,6]. In 2011, the metabolite trimethylamine N-oxide (TMAO) was first associated with CVD [7]. TMAO is formed by sequential metabolism by select members of the commensal gut microbiome and the host [7,8]. First, trimethylamine (TMA) is released from dietary choline by specific bacteria containing the *cutC/D* gene cluster, which encodes choline TMA lyase. TMA that is absorbed into the circulation is then oxidized to TMAO by hepatic flavin-containing monooxygenase 3 (FMO3) [7,8]. Other quaternary amines, such as carnitine and betaine, from exogenous (i.e., diet) or endogenous (phospholipid membranes, etc.) sources can also be converted to TMA by related bacterial TMA lyases [9,10,11,12]. Strategies to limit TMAO concentrations in circulation include reduced substrate intake, targeting bacterial TMA production, and targeting FMO3 conversion of TMA to TMAO. For various reasons, reducing bacterial production of TMA in the gut appears to be the most promising [13,14,15,16,17]. Given the lack of approved pharmacological interventions to prevent or reverse elevated TMAO concentrations or reduce bacterial TMA production, there is continuing interest in developing both drugs and lifestyle interventions (diet, etc.) to achieve this outcome to lower CVD incidence and burden. Such approaches are progressing in areas such as the development of pharmaceutical choline TMA lyase inhibitors [16,17], as well as exploration of dietary compounds that lower TMA production by various as-yet unknown mechanisms [18,19,20,21].

Given the interest in strategies to reduce TMA generation in the colon, there is a need for in vitro models of conversion of choline and other quaternary amines to TMA with suitable fidelity to in vivo conditions, sufficient throughput to meet experimental needs, and comparative ease of use. Various in vitro models have been proposed for studying choline conversion to TMA, including ex vivo fecal fermentation [16,22,23], pure bacterial culture of TMA producers such as *Proteus mirabilis* [16,17] or bacteria transfected with *cutC/D* [16,17], or non-viable lysates of such bacteria [16,17]. Finally, purified choline TMA lyases (CutC/D proteins) have been utilized [24]. We previously developed and validated a high-throughput 96-well anaerobic fecal fermentation method to study bacterial TMA production [18,19]. Using fecal inocula and isotopically labelled substrate (choline-d_9_), this method shows absolute dependence on the presence of fecal bacteria, no background interference (substrate or product), and essentially 1:1 conversion of choline-d_9_ to TMA-d_9_, with ~100% conversion in untreated controls within 12–24 h, depending upon the inoculum and other experimental parameters. We have employed this system to study the capacity of individual bioactive compounds and foods to reduce TMA production [18,19,20,21]. The advantages of this system are the use of labelled substrates and high-throughput capacity. However, there are significant drawbacks to this approach. First, this model uses reanimated fecal inocula without allowing the system to stabilize. Second, our model is a single-compartment model that neither establishes nor differentiates the distinct ecological regions of the lower gut. Finally, ours is a static batch model that does not involve a continuous flow of nutrients and fluid into, through, and then out of the system. Thus, our system is ideal for rapid, high-throughput screening to identify promising lead compounds, but a system with greater fidelity to in vivo conditions is required for further in-depth investigation of TMA formation and its inhibition for translation to animal models and human trials.

There are various rigorous and well-characterized stabilized and multi-compartment in vitro models of the colon, including the TNO in vitro model of the colon (TIM-2) [25] and Simulator of the Human Intestinal Microbial Ecosystem (SHIME^®^) [26]. These systems are fully stabilized, have regional differences, and successfully model the microbiome ecologies of the colon. The drawbacks of these models are their size, cost, and, most importantly, their very low throughput. These models are ideal for final validation of in vitro results prior to human studies, but they are not well-suited to studying the gut microbiome with multiple treatments and significant replication. We recently published validation of the MiGut model [27], a three-stage model mimicking conditions in the proximal, mid, and distal colon with continuous flow from proximal → distal (Figure 1). The model is seeded with human feces and stabilized for ~12 days prior to experiments. The system is monitored by automated data collection and tracking of resident microbial populations. The model is small and scalable: each MiGut platform has four parallel guts with three regions each. The specific parameters of the MiGut setup have been described previously [27].

To the best of our knowledge, no data exist on regional differences in *cutC/D* abundance and/or spatial–temporal TMA production in vitro or in vivo. This question is particularly challenging to address in vivo, as reliance on fecal sampling masks regional differences. This lack of understanding of potential differences in regional TMA production and differences in abundance of *cutC/D* and TMA-producing bacteria may hinder our ability to associate gut bacterial profiles with TMA and TMAO production [28]. The MiGut model represents an opportunity to perform gut microbiome modelling of the various regions of the colon with greater fidelity to in vivo conditions than our previous 96-well static model, but with greater throughput than the SHIME model. This model may be particularly useful for studying TMA production and strategies to inhibit the same, with a view to translating experimental findings to preclinical models. The objective of the present study was to characterize the spatial–temporal metabolism of choline-d_9_ to TMA-d_9_ in MiGut to demonstrate this model’s utility for studying regional differences in colonic TMA production. We hypothesized that metabolism of choline to TMA would differ between vessels (representing different colon regions) of the MiGut in vitro colon microbiome model.

## 2. Materials and Methods

### 2.1. Preparation of Fecal Slurry

Single fecal samples from healthy donors (aged > 30 years with no history of antimicrobial usage in the previous 6 months) were used to make a fecal slurry (10% *w*/*v*) using pre-reduced PBS and filtered to remove large particulate matter. Fecal samples were kept anaerobically using anaerobic sachets (Oxoid, Basingstoke, UK) inside the collection zip bag and were used within 24 h of production. Each vessel of each MiGut model was seeded with 35 mL of this fecal slurry to start the experiment. The collection and use of human feces in our gut model has been approved by the Business, Environment, Social Sciences (BESS + FREC) Ethics Committee, University of Leeds (0624—The interplay between nutrition and the gut microbiota). Participants were provided with a participant information sheet (PIS) detailing a lay summary of the in vitro gut model and the scientific work they are contributing to by providing a fecal donation. Within this PIS, it is explained that by providing the sample, the participant is giving informed consent for that sample to be used in the gut model.

### 2.2. MiGut Model Setup

MiGut models were set up as described previously [27]. Briefly, each MiGut reactor base was fitted with a lid, sampling ports, media/acid/alkaline/nitrogen connectors, and tubing, then autoclaved as a single unit prior to use. The EasyFerm Plus PHI Arc 120 pH probes (Hamilton Company, Bonaduz, Switzerland) were sterilized and calibrated prior to inserting them into the MiGut models. Each vessel is pH controlled at V1 = 5.5 (±0.1), V2 = 6.25 (±0.1), and V3 = 6.75 (±0.1), with 0.1 M HCl or 0.1 M NaOH used to maintain these pH ranges. Each MiGut reactor is temperature controlled at 36.5 °C (±0.5 °C), sparged with nitrogen (99% purity), and a media flow rate equivalent to (D = 0.015 h^−1^) to reflect in vivo colonic conditions (Figure 1A,B). Media composition is outlined in Appendix A.

### 2.3. Choline-d_9_ Fermentations

#### 2.3.1. Experiment 1 Timeline

To determine if MiGut could be used to assess the microbial metabolism of choline to TMA, 4 MiGut models were seeded with the same fecal slurry from a single donor, which assessed model reproducibility (*n* = 4 replicates). The microbial populations were allowed to stabilize for two weeks, which allowed different microbiota profiles to establish in each vessel based on the physicochemical conditions of that vessel (Figure 1A,B). At this point, the media was turned off, which kept the microbial ecologies in the three vessels separated due to the media flow restriction, but the other environmental parameters were unchanged (Figure 1C). Choline-d_9_ chloride (Cambridge Isotope Laboratories, Tewksbury, MA, USA) was added to each vessel of each model at a final concentration of 150 µM. Samples were collected at 0 (before choline-d_9_ addition), 1, 2, 3, 4, 5, 6, 8, 12, 14, 20, and 24 h after choline addition. Each sample was immediately mixed 1:1 with acetonitrile and snap frozen. We previously validated anaerobic fecal fermentations for TMA-d_9_ by demonstrating that the loss of choline-d_9_ is completely dependent on the presence of fecal inoculum, and the appearance of TMA-d_9_ is completely dependent upon the presence of both fecal inoculum and choline-d_9_ substrate [18].

#### 2.3.2. Experiment 2 Timeline

To determine if choline metabolism could be detected using a different fecal microbial ecology and to sample earlier timepoints after choline installation, we performed a further 2 MiGut models (i.e., *n* = 2 replicates) seeded with a different fecal slurry from the first experiment. Similar to the first experiment, a fecal slurry from a single donor was used to seed each vessel of 2 MiGut models, and the microbial populations were allowed to stabilize for two weeks. The media pump was stopped before the addition of choline-d_9_ chloride at a final concentration of 150 µM. Samples were collected at 0 (before choline-d_9_ addition); 5, 15, 30, and 45 min; and 1, 1.5, 2, 3, 4, 5, 6, 8, and 24 h. Each sample was immediately mixed with 1:1 with acetonitrile and snap frozen.

### 2.4. Measurement of Choline-d9 and TMA-d_9_

Choline-d_9_ and TMA-d_9_ were quantified in fermentation samples, as described previously [1]. To extract choline-d_9_, 25 μL of fermentation sample was mixed with 10 μL of ZnSO_4_ solution (5% *w*/*v* in water), 100 μL acetonitrile, and 20 μL choline-1-^13^C-1,1,2,2-d_4_ chloride [internal standard (IS) 10 μM, MilliporeSigma, Burlington, MA, USA] in 96-well plates. After sonication for 5 min in a water bath, samples were filtered through AcroprepAdv 0.2 μm WWPTFE 96-well filtering plates (Pall Corporation, Port Washington, NY, USA) by centrifugation (10 min, 3400× *g*), collected in a fresh 96-well collection plate, and frozen at −80 °C until UPLC-MS/MS analysis. TMA-d_9_ requires a derivatization process to the quaternary amine compound ethyl betaine-d_9_ to facilitate LC-MS/MS ionization. Briefly, 25 μL of fermentation sample was mixed with 20 μL of TMA-^13^C_3_-^15^N chloride IS solution (20 μM, MilliporeSigma) for derivatization of TMA-d_9_ (to ethyl betaine-d_9_) or TMA-^13^C_3_-^15^N (to ethtylbetaine-^13^C_3_-^15^N), 8 μL 32% ammonia, and 120 μL aqueous ethyl bromoacetate (20 mg/mL, MilliporeSigma), and let sit for 30 min. Then, 120 μL 50% acetonitrile/0.025% formic acid in distilled water was added. TMA-d_9_ samples were filtered and stored, as described above. Refer to Appendix A for structures and derivatization schemes.

After extraction, choline-d_9_, ethylbetaine-d_9,_ and their respective IS compounds (choline-1-^13^C-1,1,2,2-d_4_ and ethtylbetaine-^13^C_3_-^15^N) were analyzed by UPLC-ESI-MS/MS. TMA-d_9_ and TMA-^13^C_3_-^15^N were analyzed separately from choline-d_9_ and choline-1-^13^C-1,1,2,2-d_4_, but with the same UPLC-ESI-MS/MS method. Briefly, separation was achieved on a Waters Acquity UPLC system (Milford, MA, USA) with an ACQUITY BEH HILIC column (1.7 μm, 2.1 × 100 mm) coupled to a Waters ACQUITY BEH HILIC guard column (1.7 μm, 2.1 × 5 mm) (Waters). Mobile phases consisted of 5 mM ammonium formate in water (pH 3.5) (A) and acetonitrile (B). The gradient was isocratic at 80% B for 3 min, with a flow rate of 0.65 mL/min. Column temperature was 30 °C, and the autosampler was at 10 °C. Quantification was achieved with a Waters Acquity triple quadrupole mass spectrometer. Source and capillary temperatures were 150 and 400 °C, respectively. Capillary voltage was +0.60 kV, and desolvation and cone gas flows (both N_2_) were set at 800 and 20 L/h, respectively. Electrospray ionization (ESI) was operated in positive mode, and data were acquired by multiple reaction monitoring (MRM) in MS/MS mode. MRM fragmentation conditions of analytes and IS compounds can be found in Table 1.

For sample quantification, serial dilutions (0–400 µM) of choline-d_9_ and TMA-d_9_ standards were prepared to obtain external calibration curves in a relevant background matrix. Standards were then prepared by the same extraction (and derivatization) methods used for fermentation samples and analyzed by UPLC-ESI-MS/MS. Samples were quantified by interpolating the analyte/IS peak abundance ratio using the standard curves. Data acquisition was carried out using MassLynx software (V4.1, Waters).

### 2.5. Microbiome Analysis

In a separate MiGut experiment using the same fecal donor as Experiment 1, we determined the spatial microbial ecologies between the first and last vessels in the MiGut system, using taxonomic analysis via shotgun metagenomic sequencing to identify the microbial differences.

#### 2.5.1. DNA Extraction and Sequencing

Upon reaching steady state, 1 mL samples from vessels 1 and 3 were added to Lysis matrix E bead tubes (Qiagen, Manchester, UK), and the microbial cells were harvested by centrifugation at 14,000 rpm for 10 min. The DNA was extracted from these microbial pellets using the FastDNA spin kit for soil (MP Biomedicals, Derby, UK) following the manufacturer’s instructions. DNA was stored at −80 °C for downstream analysis. Metagenomic library preparation and sequencing were performed by the University of Leeds Genomics Facility. Extracted DNA was diluted to 500 ng and sheared to 200–300 bp using an E220 focused ultrasonicator (Covaris, Brighton, UK). The NEBNext Ultra DNA Library prep kit for Illumina was used for adaptor ligation and PCR enrichment, following the manufacturer’s instructions. Libraries were sequenced using an Illumina HiSeq 3000 sequencer (Cabridge, UK) at the University of Leeds.

#### 2.5.2. Metagenomic Sequence Analysis

Sequence reads underwent quality control (using FastQC; v0.11.9) before removing the adapter sequences and low-quality bases (Trimmomatic; v0.39). Forward and reverse reads were paired (PEAR; v0.96) and aligned against the NCBI non-redundant sequence database (NCBI-nr database) using DIAMOND (v2.0.8), and MEGAN6 (v6.22.2) was used for taxonomic analysis.

### 2.6. cutC Quantitative PCR

Primers N24 (AACTTAACGAGGCGCTCAAA) and N27 (AGTATGCTGGCAGAGCGAAT) were used to determine the presence of *cutC* in vessels 1, 2, and 3, as described by Wang et al. [29]. Briefly, extracted DNA was diluted to 5 ng/µL by Qubit dsDNA BR assay for use in SYBR green quantitative PCR (qPCR) (QIAGEN QuantiNOVA SYBR master mix) following the manufacturer’s instructions. The qPCR program [initial denaturation: 95 °C for 5 min; amplification: 45 cycles (95 °C 30 s, 54 °C 30 s, 72 °C 2 min)] was designed for amplification of low-abundance targets on a QTower^3^ Thermocycler (Analytik Jena, London, UK). A melt curve was used to determine specificity. Results are expressed as mean ±SD C_t_ values from 3 technical replicates on the same qPCR plate.

### 2.7. Data Analysis and Statistics

For choline-d_9_ and TMA-d_9_ kinetics data, area-under-the-curve (AUC) values were calculated for each replicate and vessel using Microsoft Excel plugins. Any negative values were converted to 0 prior to AUC calculation. AUC values were analyzed by 1-way ANOVA with Tukey’s *post hoc* test to compare group means. Within each experiment and analyte, data were analyzed by 2-way repeated measures ANOVA with these factors: vessel and time. Sphericity was not assumed. If a significant main effect of vessel or time and/or their interaction was detected, vessel means for each analyte within each time point were compared using Tukey’s *post hoc* test to correct for multiple comparisons (one family per time point). Data analysis and graphing were performed using Prism version 10.3.1 (GraphPad, La Jolla, CA, USA). Alpha was determined *a priori* for all statistical analyses as 0.05.

## 3. Results

### 3.1. Choline-d_9_ Conversion to TMA-d_9_ Differs by Simulated Colon Region

Allowing the microbiota to reach steady state within the three vessels resulted in a differentiated microbiome established between the vessels with shared and unique bacterial taxa (genera) amongst the three compartments (Appendix A). This metagenomics approach identified 70 different bacterial genera in vessel 1, 28 of which were uniquely found in vessel 1. The number of different bacterial genera increased in the other vessels, where vessel 3 supported 182 bacterial genera, of which 140 were unique to vessel 3 compared with vessel 1. Stopping the media flow, we were able to isolate these ecologies to assess the microbial choline-d_9_ conversion to TMA-d_9_ in each region of the MiGut model over 24 h. The results of Experiment 1, using *n* = 4 parallel gut models, are shown in Figure 2. Choline-d_9_ was rapidly utilized in vessels 2–3 (representing the mid and distal colon), with most choline-d_9_ used within the first 6 h, but little choline-d_9_ metabolism was observed in vessel 1 (proximal) (Figure 2A). TMA-d_9_ appearance kinetics mirrored choline-d_9_ utilization kinetics, with TMA-d_9_ rapidly reaching maximal levels in 6 h in vessels 2–3, with very little TMA-d_9_ production over 24 h in vessel 1 (Figure 2B). AUC values for choline-d_9_ and TMA-d_9_ are shown in Figure 2C,D, respectively. AUCs agree with the kinetic curves, indicating that vessel 1 had significantly greater choline-d_9_ concentrations (i.e., little utilization) and lower TMA-d_9_ production compared to vessels 2–3, which were essentially identical. The only difference between vessels 2–3 was that choline-d_9_ utilization appeared to be more rapid in vessel 3 than in vessel 2 in the first 2 h (Figure 2A), with slightly increased TMA-d_9_ appearance in vessel 3 in the first 1 h (Figure 2B).

Due to the unexpected finding that choline-d_9_ conversion to TMA-d_9_ was much lower in vessel 1 compared to vessels 2–3, we repeated the experiment in *n* = 2 gut models using a different fecal ecology (Experiment 2). An additional goal of this second experiment was to focus on the earlier time period where rapid choline-d_9_ utilization was observed in the first experiment, and so sampling was concentrated in the first 8 h. The results of Experiment 2 are shown in Figure 3. Again, choline-d_9_ conversion to TMA-d_9_ was significantly lower in vessel 1 compared to vessels 2–3 (Figure 3A,B). AUC values shown in Figure 3C,D reflected the corresponding kinetic curves. Some differences were observed between the two experiments. First, vessel 2 appeared to convert choline-d_9_ to TMA-d_9_ more rapidly compared to vessel 3 (Figure 3A,B), which is the opposite of the observations in Experiment 1, where vessel 3 was slightly more rapid than vessel 2 (Figure 2A,B). In both experiments, choline-d_9_ was completely used up in vessels 2–3 at 24 h (Figure 2A and Figure 3A), and TMA-d_9_ concentrations were similar in vessels 2–3 at 24 h (Figure 2B and Figure 3B). Choline-d_9_ AUCs did not differ between vessels 2–3 in either experiment (Figure 2C and Figure 3C), nor did TMA-d_9_ AUCs (Figure 2D and Figure 3D). The extent of choline-d_9_ metabolism to TMA-d_9_ differed between the two experiments. In Experiment 1, vessels 2–3 appeared to completely metabolize choline-d_9_ to TMA-d_9_ (Figure 2A,B), whereas conversion was less complete in Experiment 2 (Figure 3A,B). Furthermore, in Experiment 1, very little choline-d_9_ metabolism was observed in vessel 1, whereas ~50% metabolism was observed in vessel 1 during Experiment 2. Despite these differences in choline-d_9_ metabolism, TMA-d_9_ production in vessel 1 did not appear to differ between the two experiments.

Differences between vessels at each time point for each analyte are shown in Appendix A (Experiment 1: A and B, Experiment 2: C and D). Note that due to the smaller sample size (*n* = 2) in Experiment 2, fewer statistically significant differences were observed between vessels.

### 3.2. Microbiome Characterization

Given the differences in choline-d_9_ metabolism and subsequent TMA-d_9_ production between vessel 1 vs. vessels 2–3 in our MiGut system, we investigated whether we could detect *cutC* abundance. Surprisingly, we detected the presence of *cutC* in vessel 1; there was a C_t_ of 40.1, which indicates low abundance of *cutC* (consistent with the literature [30]); however, vessels 2 and 3 had even lower abundances of this gene, C_t_ of 41.1 and 44.9, respectively, compared with vessel 1 (Figure 4). This prompted us to investigate the specific microbial taxa that reside in vessel 1. The microbial ecology in vessel 1 has microbial taxa that are known to convert choline to TMA through the presence of *cutC* (Appendix A). These findings suggest that the bacteria possessing the genes required to convert choline to TMA are present in vessel 1, but that some factors, such as poor gene expression or enzyme activity, may limit this process locally.

## 4. Discussion

The results presented here in this multi-compartment in vitro model suggest that choline conversion to the proatherogenic microbial metabolite trimethylamine may primarily occur in the more distal regions of the colon. There are multiple potential explanations for the comparatively low conversion of choline-d_9_ to TMA-d_9_ in V1 (the region modelling the proximal colon) compared to V2–V3 (the regions modelling the mid and distal colon). First, the greater nutrient density in the V1 region may favor bacterial metabolism of more energy-dense substrates compared to V2–V3, where such substrates may be depleted and, thus, choline-d_9_ is more readily metabolized. Bresciani et al. showed that the presence of energy-dense nutrients such as sugars causes gut bacteria to deprioritize metabolism of quaternary amines to TMA [22]. Second, the differential ecologies along the colon may be such that the bacterial genera possessing the *cutC/D* gene cluster encoding for choline TMA lyases are present at lower relative and absolute abundance in V1. Third, genera carrying the gene cluster may be present in V1, but their metabolic activity and/or viability may be suppressed by other competitors. Fourth, gene expression and/or enzyme activity may be suppressed for some reason in V1. Differences in choline-d_9_ metabolism to TMA-d_9_ between vessels (colon regions) may be due to differences in *cutC/D* abundance and expression, or abundance/viability of bacteria that carry *cutC/D*, between experiments [9,28,31,32]. Additionally, other gene clusters are known to catalyze choline to TMA, such as carnitine oxygenase (*cntA*) or betaine reductase (*grdH*) [33], which have not been investigated here. Furthermore, variations in the presence of bacteria [34] or methanogenic Archaea [35] that utilize choline (but do not produce TMA), or which utilize TMA [36], may also be a plausible explanation. Here, microbes can utilize choline to produce other metabolites, such as betaine and phosphatidylcholine. Thus, choline utilization does not always equate to TMA production, and regional microbiome differences between individuals will affect TMA production. These questions can be addressed initially through the novel, high-throughput capabilities of the MiGut platform and are logical next steps in this line of investigation.

Most importantly, this intriguing finding of regional differences in TMA production raises the critical question of whether the same phenomenon is observed in animals and humans in vivo. To our knowledge, there are no data in the literature on regional differences in TMA production in vivo. Experiments to study regional differences in metabolism are currently nearly impossible in humans. However, multiple approaches could be used in rodent models. The colon could be exteriorized and maintained in warm anaerobic (reduced) saline in live but sedated animals, facilitating direct micro sampling over time with tuberculine syringes from various regions following oral gavage of choline-d_9_ (or direct introduction into the ileum or proximal colon). Alternatively, animals could be euthanized, and the ileum and colon could be immediately excised in an anaerobic chamber, the regions separated, and their contents and/or mucosa extracted and exposed to choline-d_9_ with subsequent sampling over time. Finally, a large number of rodents could be gavaged with choline-d_9_, and cohorts euthanized over time for regional sampling, an approach we have used previously to characterize regional phenolic metabolism by the microbiome [37], but this has ethical implications. Regional differences are challenging to assess in humans, given the invasive nature of sampling various colon regions. However, one approach could be to administer labelled choline-d_9_ along with a mixed sugar probe (to track gut permeability) [38,39] and track the appearance of TMA-d_9_ and choline-d_9_, along with the region-specific sugar probes to determine where TMA-d_9_ is being formed. However, the selectivity for these probes to different colonic regions is not well-defined.

The implications of maximal choline TMA lyase activity occurring in the more distal regions of the colon are physiologically significant. First, there may be many species that carry *cutC/D* (encoding choline TMA lyase) but which reside or are active primarily in specific gut regions. An understanding of where TMA lyase action occurs may allow us to focus on specific *cutC/D*-carrying bacteria that reside in the high activity regions, with less focus on those that carry the gene but are primarily in low activity regions. Second, if the conditions that suppress TMA production in the proximal colon can be identified, interventions such as diet, prebiotics, probiotics, or pharmaceuticals may be employed to alter conditions in more distal regions to become less favorable to TMA lyase expression and activity. This may include alteration of pH via prebiotics and probiotics that produce short-chain fatty acids and acidify the lumen, selective promotion of suppressor bacterial species, etc. Third, targeted delivery strategies could be employed to precisely deliver TMA lyase inhibitors such as 3,3-dimethyl-1-butanol (DMB) [17,40] or dietary phenolics [18,19], specifically to the needed regions to avoid dilution or degradation of the inhibitors in regions with low TMA lyase activity. Such delivery strategies may include pH-responsive polymers (as we have previously reported for phenolics and antibiotics [41,42]), encapsulation of inhibitors with fermentable material whose degradation and thus release rate can be controlled, etc. One such example of interest to us is dietary phenolic compounds. These compounds are metabolized by commensal gut bacteria to smaller compounds [37,43]. We have performed some initial work to identify the comparative TMA-lowering properties of native phenolics in food and their microbial metabolites [18,19,20,21]. Targeted delivery could be designed to protect the native food phenolics from bacterial metabolism until they reach the high TMA lyase activity regions. Conversely, postbiotics (preformed microbial metabolites) could also be delivered similarly if these metabolites are found to be the most effective at lowering TMA production.

Finally, an understanding of regional differences in microbial community composition and associated TMA lyase activity may help us understand *cutC/D* as a marker of TMA production capacity and CVD risk. Fecal samples are the only readily available samples from humans from which the gut microbiome can be studied. However, feces are a composite of living, inactive, and dead cells from across all regions of the gut. Studies have indicated that fecal composition and *cutC/D* copy numbers are poor predictors of TMA/TMAO production and disease risk due to differences in metabolic activity, viability, and gene expression (potentially confounded by the fact that each region cannot be studied individually in humans) [28], although there remains some disagreement on this point [32].

This work has limitations. Firstly, our MiGut model is a microbiome model and does not replicate the complexity and regional differences of host cellular responses of the in vivo human gut and its associated metabolome. Host factors can tune the microbial ecology and, thus, their functionality, the effects of which are absent in MiGut and may shape choline metabolism. Furthermore, the lack of metabolite absorption results in an increased accumulation in MiGut, which could affect the microbial metabolic pathways and influence choline metabolism. While provocative, the present finding must eventually be confirmed in vivo. Second, any in vitro gut microbiome model is inherently limited by the human fecal samples used to colonize it. The present observations may not be generalizable to humans. Further studies are needed, using samples from more diverse donors to validate these findings. This weakness is somewhat mitigated by the fact that different fecal donors were used for Experiments 1 and 2, with similar results. Another limitation is the *cutC* primers used. Although these are degenerate primers that have been used before [29], more comprehensive degenerate primers may provide better amplification of a broader swath of *cutC* from multiple species [44]. Finally, other quaternary amines from the diet (carnitine and betaine) can be metabolized into pro-atherogenic TMA [11,12], and the regional dependence of these pathways remains unknown. More replications are also needed to confirm these findings, including the use of additional donors to assess interindividual differences.

As outlined above, additional studies are needed to validate and expand upon this finding. In the near term, additional studies using different donors must be performed to further confirm our finding. The next iteration of experiments will then be “full-flow” studies in MiGut where choline-d_9_ is added only to V1 and media flow is allowed from V1→V3 for enhanced physiological relevance (similar to the setup in Figure 1B). Another immediate priority will be to fully characterize the abundance and metabolic activity of *cutC/D*-bearing bacteria, *cutC/D* gene copy number, and *cutC/D* expression in V1–V3 to understand how these factors may influence TMA production. Moving forward, in-depth taxonomic and function characterization of TMA production by *cutC/D* and other TMA lyase-encoding gene clusters will be required using metagenomics, metatranscriptomics, and targeted qPCR to fully understand these spatial–temporal differences. Finally, initial in vivo confirmatory experiments in rodents are a priority.

The central finding of the present study is that choline metabolism to TMA by the commensal gut microbiome varies by region in an in vitro model that closely mimics regional differences in the resident microbiome profiles in vivo. Furthermore, preliminary evidence suggests that *cutC* levels do not correlate with TMA production. Given the known challenges of correlating fecal microbiome profiles with blood TMA/TMAO levels [28], our data suggest a provocative explanation that *cutC/D* expression and/or activity differ between various colonic regions. Further investigation to unravel the mechanisms behind this regional specificity of choline conversion to TMA may reveal keys to controlling gut TMA production (and, by extension, blood TMAO concentrations). These mechanisms can then be targeted and exploited using diet and or pharmacological strategies targeting the regions of TMA production to control blood TMAO concentrations and CVD risk.

## 5. Conclusions

In conclusion, the present study reports the first known finding that choline conversion to the pro-atherogenic metabolite TMA by human gut bacteria may vary by colonic region. This finding has implications for understanding TMA production in vivo, as well as the design of strategies to control TMA (and TMAO) levels with the end goal of reducing long-term CVD risk. This discovery warrants further study in vitro and in vivo.

## Figures and Tables

**Figure 1 metabolites-15-00552-f001:**
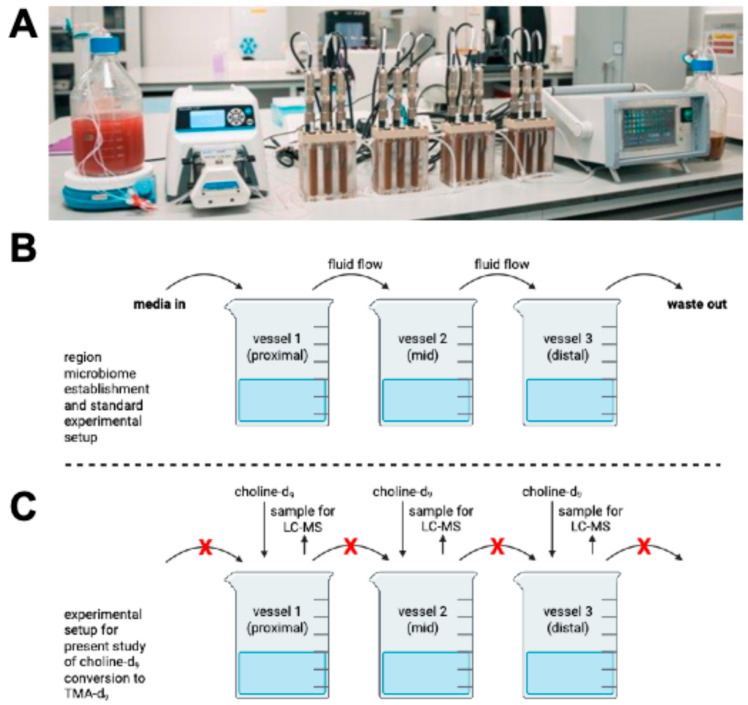
MiGut setup (**A**). A single human MiGut in vitro platform consisting of 4 independent triple-staged models. All environmental parameters are measured and controlled by the controller unit on the right and fed with the media outlined in Appendix A. Schematic showing model regional microbiome establishment with typical experimental setup with flow between vessels (**B**) vs. setup for current experiment with no flow between vessels (**C**).

**Figure 2 metabolites-15-00552-f002:**
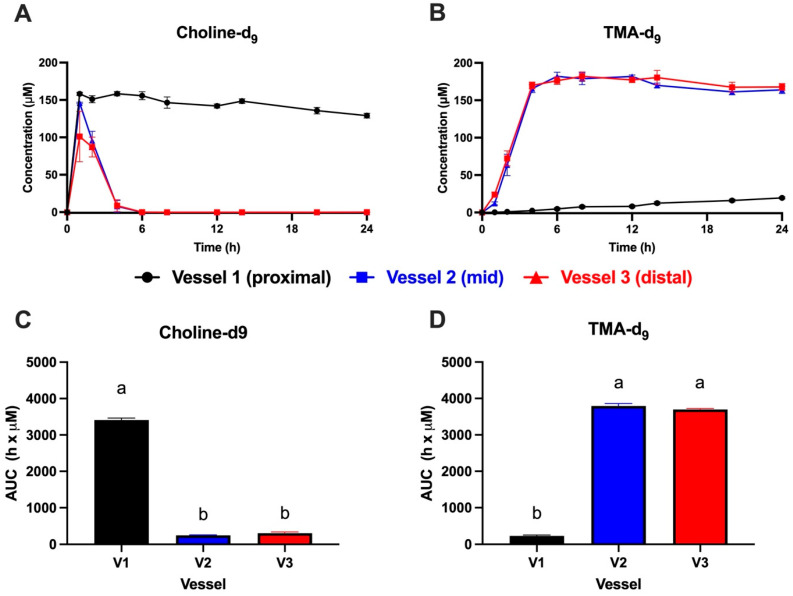
Kinetics of choline-d_9_ metabolism (**A**) and trimethylamine (TMA)-d_9_ production (**B**) with corresponding area-under-the-curve (AUC) values for choline-d_9_ (**C**) and TMA-d_9_ (**D**) in the MiGut model following the addition of 150 μM choline-d_9_ in each of the 3 vessels, with no flow between vessels (Experiment 1). Values represent the mean ± SEM from *n* = 4 parallel gut models (i.e., 4 replicates per vessel). AUC values were calculated for each replicate and vessel separately. Any negative values were converted to 0 prior to AUC calculation. AUC values were analyzed by 1-way ANOVA with Tukey’s *post hoc* test to compare group means. Bars not sharing a common superscript letter are significantly different (*p* < 0.05). V1: vessel 1, V2: vessel 2, V3: vessel 3.

**Figure 3 metabolites-15-00552-f003:**
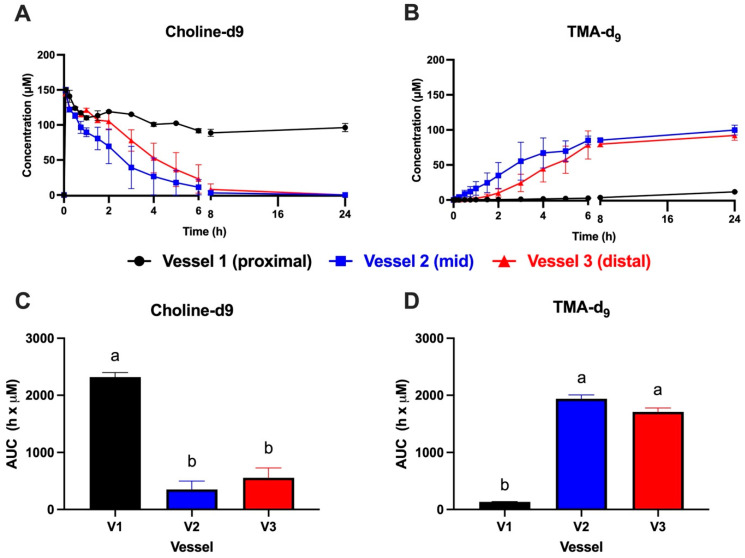
Kinetics of choline-d_9_ metabolism (**A**) and trimethylamine (TMA)-d_9_ production (**B**) with corresponding area-under-the-curve (AUC) values for choline-d_9_ (**C**) and TMA-d_9_ (**D**) in the MiGut model following the addition of 150 μM choline-d_9_ in each of the 3 vessels, with no flow between vessels (Experiment 2). Values represent the mean ± SEM from *n* = 2 parallel gut models (i.e., 2 replicates per vessel). AUC values were calculated for each replicate and vessel separately. Any negative values were converted to 0 prior to AUC calculation. AUC values were analyzed by 1-way ANOVA with Tukey’s *post hoc* test to compare group means. Bars not sharing a common superscript letter are significantly different (*p* < 0.05). V1: vessel 1, V2: vessel 2, V3: vessel 3.

**Figure 4 metabolites-15-00552-f004:**
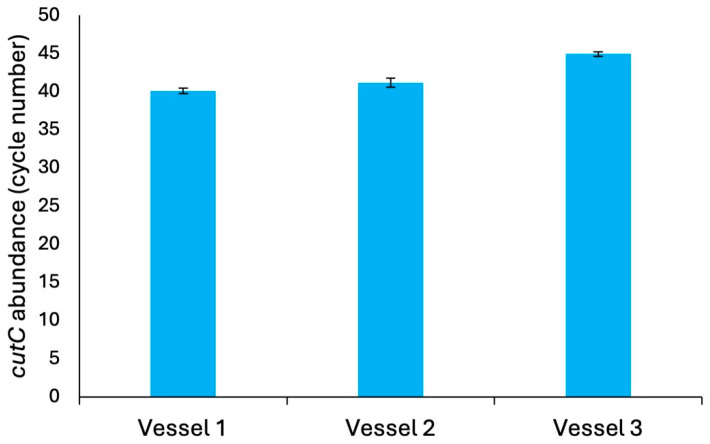
Abundance of the choline metabolism gene, *cutC*, in vessels 1, 2, and 3 of MiGut, as determined by qPCR. Results expressed as mean cycle number (±SD) from 3 technical replicates on the same qPCR run.

**Table 1 metabolites-15-00552-t001:** Multi-reaction monitoring parameters for the detection of choline-d_9_, TMA-d_9_, and their internal standards.

Compound	MW	MS/MS Transition	CV (V)	CE (eV)
Choline-d_9_	113.2	113.3 > 69.1	40	16
Choline-1-^13^C-1,1,2,2-d_4_	109.2	109.3 > 60.3	36	18
Ethyl betaine-d_9_ ^a^	155.2	155.3 > 127.2	34	20
Ethyl betaine-^13^C_3_-^15^N ^a^	150.2	150.3 > 122.2	34	18

^a^ Abbreviations: TMA, trimethylamine; MW, molecular weight; CV, cone voltage; and CE, collision energy. aTMA derivatives for ionization.

## Data Availability

The original contributions presented in this study are included in the article/Appendix A. Further inquiries can be directed to the corresponding author.

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
