# Peer review of "Choline Metabolism to the Proatherogenic Metabolite Trimethylamine Occurs Primarily in the Distal Colon Microbiome In Vitro"

_metabolites, 2025, doi:10.3390/metabo15080552_

Round 1

Reviewer 1 Report

Comments and Suggestions for Authors

Thank you for submitting your manuscript investigating regional differences in choline metabolism to TMA using the 3-compartment MiGut in vitro colon model. Your work addresses a critical gap in understanding how gut microbial activity varies across the colon and its implications for cardiovascular disease risk through TMA/TMAO pathways. The study is innovative and timely, and the use of isotopically labelled choline is a strength. 

Reviewer 2 Report

Comments and Suggestions for Authors

The manuscript entitled “Choline Metabolism to the Proatherogenic Metabolite Trimethylamine Occurs Primarily in the Distal Regions of the MiGut 3-Compartment In Vitro Colon Microbiome Model” presents novel findings on the spatial dynamics of choline metabolism within the human colon using a sophisticated in vitro model.

Below are some comments and inputs:

  • Vessel 1 showed higher cutC abundance, yet little TMA-d9 was produced. This contradiction is acknowledged, but further insights into microbial viability, gene expression, or competing metabolic pathways (e.g., alternative substrate use, TMA consumption) would improve the explanation.

  • The major drawback of this study is the small size number in terms of donors. Although two different donors were used, the sample size remains small and limits generalizability. The authors should discuss this limitation more explicitly and suggest how future studies might address interindividual variability.

  • Again, I would suggest to increase the size number in Experiment 2 which included only two MiGut units. the small sample size reduces confidence in statistical comparisons. This should be acknowledged more transparently in the result section and limitations section.

  • Given the findings, exploring whether microbial gene expression (e.g., mRNA of cutC) differs across regions would enhance mechanistic depth. The authors suggest this for future work, but a brief qPCR for expression rather than abundance could have strengthened conclusions.

  • It is unclear whether negative controls (e.g., heat-inactivated microbiota or sterile media) were used to rule out non-microbial conversion. Clarifying this would strengthen the validity of the microbial specificity claim.

  • Terms such as "TMA-d9" and "choline-d9" should be consistently formatted throughout.

Reviewer 3 Report

Comments and Suggestions for Authors

Thank you for the opportunity to review the manuscript entitled "Choline Metabolism to the Proatherogenic Metabolite Trimethylamine Occurs Primarily in the Distal Regions of the MiGut 3-Compartment In Vitro Colon Microbiome Model." The work presents a timely and well-executed investigation into the spatial dynamics of choline metabolism in the gut microbiome, with important implications for cardiovascular health.

The different strengths of this work are:

  1. Innovative Model Use: The MiGut 3-compartment model is a sophisticated and appropriate tool for studying spatial microbial metabolism. Its application here is both novel and well-justified.

  2. Relevance to Human Health: The focus on TMA production has direct implications for understanding cardiovascular disease risk, making the study highly relevant to both microbiome and clinical research communities.

  3. Clear Experimental Design: The authors provide a logical and reproducible experimental setup, with appropriate controls and time points.

  4. Data Interpretation: The results are interpreted thoughtfully, with a clear link between microbial composition and metabolic output across compartments.

Below are my comments and suggestions intended to strengthen your manuscript:

Major Comments:

  1. Statistical Analysis: Please provide more detail on the statistical methods used, including effect sizes and confidence intervals where appropriate. This will improve the transparency and reproducibility of your findings.

  2. Mechanistic Insight: Consider expanding the discussion on the microbial pathways involved in choline metabolism and how these may differ regionally. This would help contextualize your findings within the broader literature.

  3. Model Limitations: A more explicit discussion of the limitations of the MiGut in vitro model—such as the absence of host-microbe interactions and immune modulation—would provide a more balanced interpretation of the results.

  4. Microbial Community Characterization: While the study mentions microbial composition, a more detailed taxonomic and/or functional analysis (e.g., metagenomics, metatranscriptomics, or targeted qPCR) would enhance the mechanistic understanding of TMA production across compartments.

Minor Comments:

  • The title, while descriptive, could be streamlined for clarity and impact.
  • Some figures would benefit from larger font sizes and more detailed legends.
  • A graphical abstract summarizing the experimental design and key findings would enhance accessibility for a broader audience.
  • The Supplemental figure 1, where the MiGut setup is shown, should be added in the main text since this is important for the understanding of the experiment section.

Conclusion:

The study is scientifically sound and offers novel insights into the spatial aspects of microbial choline metabolism. With minor revisions to enhance clarity and depth, this work will make a valuable contribution to the field.

Thank you for allowing me to review this work and I look forward to seeing the revised version.

Round 2

Reviewer 2 Report

Comments and Suggestions for Authors

well revision